# Risks and Benefits of Judo Training for Middle-Aged and Older People: A Systematic Review

**DOI:** 10.3390/sports11030068

**Published:** 2023-03-14

**Authors:** Federico Palumbo, Simone Ciaccioni, Flavia Guidotti, Roberta Forte, Attilio Sacripanti, Laura Capranica, Antonio Tessitore

**Affiliations:** 1Department of Movement, Human and Health Sciences, University of Rome “Foro Italico”, 00135 Rome, Italysimoneciaccioni@yahoo.it (S.C.);; 2International Judo Federation Academy Foundation, XBX 1421 Ta’ Xbiex, Malta

**Keywords:** judoka, martial arts, combat sports, older individuals, coaches, successful aging

## Abstract

This systematic overview aimed to review studies investigating the benefits and risks of judo training in older people, and to explore practical methodological applications (Registration ID: CRD42021274825). Searches of EBSCOhost, ISI-WoS, and Scopus databases, with no time restriction up to December 2022, resulted in 23 records meeting the inclusion criteria. A quality assessment was performed through the following tools: ROBINS-I for 10 experimental studies, NIH for 7 observational studies, and AGREE-II for 6 methodological studies. A serious risk of bias emerged for 70% of the experimental studies, whereas 100% of the observational and 67% of the methodological studies presented a “fair” quality. When involving 1392 participants (63 ± 12 years; females: 47%), the studies investigated novice (*n* = 13), amateur/intermediate (*n* = 4), expert (*n* = 4), and unknown (*n* = 3) level judoka by means of device-based, self-reported, and visual evaluation measures. Mean training encompassed 2 ± 1 sessions. week^−1^ of 61 ± 17 min for 7 ± 6 months. In relation to judo training exposure and outcomes, three main themes emerged: (i) health (56% of studies; e.g., bones, anthropometry, quality of life); (ii) functional fitness (43%; e.g., balance, strength, walking speed); and iii) psychosocial aspects (43%; e.g., fear of falling, cognition, self-efficacy). Although the included studies presented relevant methodological weaknesses, the data support the positive effects of judo training with advancing age. Future research is needed to help coaches plan judo programs for older people.

## 1. Introduction

Human longevity has increased considerably in western countries, urging policymakers to develop strategies for promoting health-enhancing active lifestyles for older people [1,2,3,4]. Ageing is a complex phenomenon characterised by a loss of physiological, functional, and cognitive capacities [5,6,7]. Whilst sedentary behaviours determine physical and cognitive declines, increase in chronic diseases, and mortality [8], active lifestyles promote behavioural, physiological, psychological, and social wellness, functional capacities, and fall prevention with advancing age [1,9,10,11,12]. Therefore, the World Health Organization (WHO) recommends older individuals to engage >2 days weekly in a multicomponent physical activity, encompassing aerobic and resistance training involving major muscle groups [5]. Furthermore, training programs for older people should combine exercise and cognition (e.g., dual tasks) [7,13]. In this framework, sport-based interventions proved to be particularly effective [3,12,14,15,16], with master athletes considered good examples of successful aging for preserving high levels of fitness, performances, mental health, social interactions, and quality of life, as well as experiencing low levels of stress, negative feelings, and depression [3,17,18,19,20,21,22].

In multimodal exercise programs encompassing postural control, coordination, mental exercises, and self-defence techniques implemented in group settings, martial arts are considered suitable for older people [5,23,24,25]. Among martial arts, Judo (jū = gentle; dō = way) is a Japanese traditional form of physical education based on the principles of “maximum efficient use of mind and body” and “mutual welfare and benefit”; it has been an Olympic sport since 1964 for men and 1988 for women [26]. Judo aims to teach practitioners an effective self-defence method and to nurture a sustainable lifestyle, promoting self-control and discipline throughout long-term physical, mental, and spiritual process, with coaches considered a life-long guide [26]. Judo is an inclusive sport promoting active physical behaviours, quality of life, physical and mental well-being, and safe fall skills in relation to the athlete’s age, sex, and level of expertise [12,26,27]. Therefore, judo has been recommended to maintain functional fitness, physiological and psychological well-being, to prevent fall-related injuries, and to sharpen gait performance by exercising relaxation, functional coordination, and self-confidence [12,24,25,28]. Despite two reviews which summarised the health outcomes of older judoka [24,29], judo is not yet included in the list of the WHOs suggested activities to prevent falls [5]. Furthermore, there is a lack of information on judo training programs for older practitioners and their effects in relation to the athletic characteristics (e.g., novice, intermediate, expert) of the judoka. Thus, to cover this knowledge gap in the understanding of physical, mental, and social effects and correlates of judo training in older practitioners, there is a need to collect comprehensive information from both theoretical and practical perspectives to guide the implementation of effective and safe training programs for older people. Therefore, the main purpose of this systematic review was to summarise evidence on the potential benefits and risks that judo training could elicit in middle-aged and older practitioners. In particular, cross-sectional, experimental, and methodological study designs will be considered. It is hypothesised that this novel information will provide guidance for practical and methodological applications for effective judo programs for older judoka.

## 2. Methods

The study is part of the co-funded European Erasmus + Sport the “EDucating Judo Coaches for Older practitioners” project (EdJCO, 622155-EPP-1-2020-1-IT-SPO-SCP)”.

### 2.1. Protocol and Registration

Registered on PROSPERO (ID: CRD42021274825), the present systematic literature review was based on the Preferred Reporting Items for Systematic Reviews and Meta-analyses (PRISMA) guidelines [30].

### 2.2. Eligibility Criteria

The inclusion criteria for the selection of studies encompassed: (i) original peer-reviewed articles published without a time restriction up to November 2022 in languages accessible to the authors (e.g., English, French, Italian, Spanish); (ii) studies involving middle-aged (i.e., ≥45–65 years) and older (i.e., ≥65 years) practitioners [31]; (iii) effects or correlates of judo training; and (iv) experimental (e.g., randomised or non-randomised controlled trials), observational (e.g., cross-sectional and longitudinal), and methodological (e.g., research notes) study designs. Conversely, exclusion criteria encompassed: (i) the focus on the effects or correlates of other sports; (ii) review articles (e.g., meta-analyses, systematic reviews); and (iii) non-peer-reviewed publications (e.g., letters to the editor, translations, book reviews).

### 2.3. Information Search and Study Selection Process

In November 2021, a systematic literature search of original articles was performed on the SPORTDiscus (EBSCOhost), PsycArticles, Institute for Scientific Information Web of Science (ISI WoS), and Scopus database. The used search string was: (master OR senior OR veteran OR old OR aging OR elderly OR adult * OR man OR men OR woman OR women OR aged OR old OR successful aging) AND (judoka OR judo OR judoist OR martial art * OR combat sport *) AND (fall * OR slip OR drop OR training OR performance). The asterisks (*) were used to pull all derivations of the similar root word (i.e., old adult * = old adult and old adults). To ensure the inclusion of the most updated articles, alert notifications of new publications were activated until December 2022. To ensure conformity with the inclusion criteria, two authors, who specialised in sports sciences and judo (i.e., certified kinesiologists and 4th dan black belt), performed independently the screening and quality assessment of pertinent studies and the data extraction. In case of disagreement on eligibility, a third author’s opinion was sought. The retrieved studies were screened based on title, abstract, and the full text. To provide a comprehensive search of the relevant articles, the snowball technique was applied.

### 2.4. Data Extraction and Synthesis

The data extraction of the final studies sample was based on the PICOS approach [30]. The information extrapolated and examined encompassed: author(s); publication year; journal; language; country in which the study was conducted; context (e.g., physical, social, environmental context, and delivery mode of physical activity); guidance (e.g., physical activity teacher, instructor); number, age range, and sex of participants included in the experimental and control groups; expertise and competitive level; training load (e.g., volume, intensity, weekly frequency); study type; dependent variable(s); confounders; main outcomes; outcomes categories; risks; and benefits. According to the PRISMA guidelines [30], data were processed through a qualitative synthesis. The effects and correlates of judo training were considered in relation to the participants’ chronological age and activity group (i.e., sedentary, judoka, sportive controls). Then, detailed tables and figures reporting the major characteristics and findings of the selected studies were created.

### 2.5. Quality Assessment

The Risk of Bias In Non-Randomised Studies of Interventions (ROBINS-I) tool [32], the Quality Assessment Tool for Observational Cohort and Cross-Sectional Studies of the National Institutes of Health (NIH) [33], and the modified Advancing Guideline Development, Reporting and Evaluation in healthcare (AGREE II) tool [34] were used to assess the quality of non-randomised controlled trials, cross-sectional, and methodological studies, respectively (Table 1 and Appendix A).

### 2.6. Non-Randomised Controlled Trials

To identify potential biases, ROBINS-I is based on the premise that an interventional study without randomisation should be compared to a hypothetical randomised controlled trial. The ROBINS-I tool contains 7 domains, divided in sub-items depending on the closed-ended answer. The first two domains address the issues to be compared before the start of the interventions (“baseline”): (i) bias to confounding, and (ii) bias in the selection of participants for the study. The third domain concerns the classification of the interventions themselves: (iii) bias in the measurement of intervention. The other domains address issues after the start of the interventions: (iv) bias to departures from intended interventions; (v) bias due to missing data; (vi) bias in measurement of outcomes; (vii) and bias in the selection of reported results.

### 2.7. Cross-Sectional Studies

To assess the methodological quality, the NIH tool consists of 14 items to appraise the following issues: (i) the research question; (ii) study population; (iii) participation rate of eligible persons; (iv) groups recruited from the same population and uniform eligibility criteria; (v) sample size justification; (vi) exposure assessed before outcome measurement; (vii) timeline of exposure and outcome assessment; (viii) different levels of the exposure of interest; (ix) exposure measures and assessment; (x) repeated exposure assessment; (xi) outcome measures; (xii) “blinding” of outcome assessors; (xiii) follow-up rate; and (xiv) confounders and statistical analyses. Thus, a “good”, “fair”, or “poor” judgment was assigned to each included manuscript based on the NIH results.

### 2.8. Methodological Studies

Previously developed to assess the methodological quality of clinical practice guidelines, the AGREE II assessment tool is a validated questionnaire comprising 23 items within 6 domains: (i) scope and purpose (*n* = 3 items); (ii) stakeholder involvement (*n* = 4); (iii) rigor of development (*n* = 7); (iv) clarity of presentation (*n* = 4); (v) applicability (*n* = 3); and editorial independence (*n* = 2). To meet the specificity of the present study aims and to contextualise the tool to the sport setting, the AGREE II items were modified replacing the following words: “clinical” with “study”, and “patients” with “participants”.

## 3. Results

### 3.1. Study Selection and Data Collection

Figure 1 shows the PRISMA flow diagram of the article selection process based on the title, the abstract, and the full text, reporting reasons for exclusion. The electronic search strategy identified 2509 contributions, with 20 articles retained after the screening for the title, abstract, and full text, further implemented with 3 additional contributions identified through the database alert notifications (Figure 1). The final list of 23 included manuscripts is shown in Table 1 and Table 2, accordingly to their publication year.

### 3.2. Study Characteristics

Regarding the geographical distribution of the studies (Table 1), 13 countries were represented, with the highest frequency of occurrence in Europe (Spain: *n* = 9 studies; Italy: *n* = 3; Poland: *n* = 3; Belgium = 1; France: *n* = 1; Ireland: *n* = 1; Serbia: *n* = 1; Sweden: *n* = 1), followed by South America (Brazil: *n* = 4), and Asia (Japan: *n* = 1). Since the first study addressing the effects of judo training on the bone mineral density, balance, and quality of life in postmenopausal women published in 2012, the interest towards the study of positive effects/risks of judo training for middle aged and older people increased (2012–2017: *n* = 11 studies, 2018–2022: *n* = 12 studies), with a peak in 2020 (*n* = 5). Three types of study design emerged: (i) experimental (*n* = 10 non-randomised controlled trials: 43.5%) [35,36,37,38,39,40,41,42,43,44]; (ii) observational (*n* = 7 cross-sectional studies: 30.4%) [45,46,47,48,49,50,51]; and (iii) methodological (*n* = 6 research notes: 26.1%) [52,53,54,55,56,57]. Specific information on judo training emerged in 13 studies only, reporting a duration of 7 ± 6 (range = 2–24) months, a weekly training volume was 2 ± 1 (range = 1–3) sessions, with each session lasting 61 ± 17 (range = 45–120) minutes.

**Table 1 sports-11-00068-t001:** Country, research area, design, and quality assessment of the studies included in the systematic literature review.

Author (Year)	Country	Research Area ^a^	Study Design ^b^	Quality Assessment	
				Tool	Overall Evaluation Rating
Borba-Pinheiro et al. (2012) [35]	Brazil	Fit/H	CT	ROBINS-I	Serious
Borba-Pinheiro et al. (2013a) [36]	Brazil	H	CT	ROBINS-I	Serious
Borba-Pinheiro et al. (2013b) [52]	Brazil	Fit/H	MT	AGREE II	Good
Lefevre et al. (2013) [45]	France	H	CS	NIH	Fair
Krampe et al. (2014) [46]	Belgium-Ireland	Fit/Psy	CS	NIH	Fair
Michnik et al. (2014) [47]	Poland	Fit	CS	NIH	Fair
Muinos et al. (2014) [48]	Spain	Psy	CS	NIH	Fair
Campos-Mesa et al. (2015) [53]	Spain	H	MT	AGREE II	Fair
Muinos et al. (2015) [49]	Spain	Psy	CS	NIH	Fair
Suarez-Cadenas et al. (2016) [50]	Serbia-Spain-Poland	Psy	CS	NIH	Fair
Del Castillo-Andrés et al. (2016) [54]	Spain	H	MT	AGREE II	Good
Del Castillo-Andres et al. (2018) [55]	Spain	H	MT	AGREE II	Fair
Toronjo-Hornillo et al. (2018) [37]	Spain	Psy	CT	ROBINS-I	Serious
Ciaccioni et al. (2019) [38]	Italy	Fit/H/Psy	CT	ROBINS-I	Moderate
Arkkukangas et al. (2020) [39]	Sweden	Fit/Psy	CT	ROBINS-I	Moderate
Borba-Pinheiro et al. (2020) [56]	Brazil	H	MT	AGREE II	Good
Campos-Mesa et al. (2020) [40]	Spain	Fit/Psy	CT	ROBINS-I	Serious
Ciaccioni et al. (2020) [41]	Italy	Fit	CT	ROBINS-I	Moderate
Franchini et al. (2020) [51]	Brazil	Fit/H	CS	NIH	Fair
Ciaccioni et al. (2021) [42]	Italy	Fit/Psy	CT	ROBINS-I	Moderate
Toronjo-Hornillo et al. (2021) [57]	Spain	H	MT	AGREE II	Fair
Sakuyama et al. (2021) [43]	Japan	H	CT	ROBINS-I	Moderate
Kujach (2022) [44]	Poland	H/Psy	CT	ROBINS-I	Serious

^a^ Study design: CT = non-randomised controlled trial/quasi-experimental studies; CS = observational cross-sectional studies; MT = methodological studies; ^b^ research area: Fit = fitness; H = health; Psy = psychosocial.

### 3.3. Participant Characteristics

Overall, the studies included 1392 participants; 1180 were assigned to training groups and 232 to control groups (Table 2). Seven studies lacked a control group [37,42,43,45,47,50,51]. The highest number of participants were included in a study from Brazil (*n* = 546) [51] and Belgium (*n* = 143) [46], respectively. The age of judo practitioners was 64.8 ± 8.5 yr., ranging from 48.9 yr. [50] to 88 yr. [39]. The relative picture for the control group was 61.1 ± 16 yr., ranging from 54 yr. to 88 yr. [39]. In total, 4 studies focused on female participants only, 5 studies included only male participants, and 12 studies presented a female representation ranging between 14 and 86%. One study [45] did not report sex-related data. In general, the studies included healthy participants, three studies included women with a diagnosed arthrosis [35,36] and judoka who underwent a surgery [45], and one study [39] also included a group encompassing a convenience sample of patients from a local healthcare centre. Regarding the judo background of the participants, 13 studies focused on novices, 4 on high-level judoka, 4 on amateur- or intermediate-level judoka, and 4 studies did not report this information.

**Table 2 sports-11-00068-t002:** Information on participants, judo training, variables, measurements, and outcomes reported in the included studies.

Author (Year)	Participants ^a^				Judo Training Load (*n*) ^b^	Variables ^c^	Measurement ^d^	Outcome ^e^
	Age (Years)	Sex (%)	Level	Health Status				
Borba-Pinheiro et al. (2012) [35]	J: 52.2 ± 5.3; C: 53.8 ± 4.4	F = 100%; M = 0%	J: Novices	Healthy	12; 3; 60	BMD	DXA	+
						Osteoporosis	OPAQ	+
						Balance	Performance test	+
Borba-Pinheiro et al. (2013a) [36]	J: 52.2 ± 5.3; C: 53.8 ± 4.4	F = 100%; M = 0%	J: 2 years of judo practice	Healthy	24; 3; 60	BMD	DXA	+
Borba-Pinheiro et al. (2013b) [52]	Over 50 years old	F = 100%; M = 0%	J: Novices	Healthy	NR	NA	NA	NA
Lefevre et al. (2013) [45]	J: 75.3 ± 8.1 (at study time); J: 70 ± 21.1 (at surgery time)	NR	Regular weekly training and competitions	Clinical	NR; 2–3; NR	Rate of return to judo after joint replacement	Ad hoc Survey	+
Kramp et al. (2014) [46]	M: 61.9 ± 4.7; S: 63.7 ± 5.0; C: 61.9 ± 4.7	F = 0%; M = 100%	M: Experts	Healthy	NR; 2–3; 90–120	Postural control	Force plate	?
						Cognition	Reaction time digit test	?
Michnik et al. (2014) [47]	J1: 65 ± 0; J2: 24 ± 0	F = 0%; M = 100%	J1: Experts; J2: Amateurs	Healthy	NR	Balance	Inertial sensors	+
Muinos et al. (2014) [48]	J: 64.1 ± 3.6 (range = 56–67)	F = 0%; M = 100%	J: Experts	Healthy	NR	Visual Acuity	Computerised Reaction time test	+
Campos-Mesa et al. (2015) [53]	Older people	NA	J: Novices	Healthy	2; 2; 60	NA	NA	NA
Muinos et al. (2015) [49]	J: 64.1 ± 3.6 (range = 56–67)	F = 0%; M = 100%	NR	Healthy	NR	Visual Acuity	Computerised reaction time test	+
Suarez-Cadenas et al. (2016) [50]	J: 48.9 ± 9.6	F = 15.2%; M = 84.8%	J: Experts	Healthy	NR	Mental toughness	SMTQ	+
						Perfectionism	Sport-MPS2	+
DelCastillo-Andrés et al. (2016) [54]	J: >55	NA	J: Novices	Healthy	9; 2; 50	NA	NA	NA
DelCastillo-Andres et al. (2018) [55]	J: >65	NA	J: Novices	Healthy	NR	NA	NA	NA
Toronjo-Hornillo et al. (2018) [37]	J: 71.5 ± 8	F = 100%; M = 0%	J: Novices	Healthy	2; 2; 60	Fear of falling	FES-I	+
Ciaccioni et al. (2019) [38]	J: 69.3 ± 3.9; C: 70.1 ± 4.5	J: F = 50%; M = 50%; C: F = 36%; M = 74%	J: Novices	Healthy	4; 2; 60	Fear of falling	FES-I	+
						Flexibility, strength, coordination	Rikli and Jones test battery	+
						Anthropometry	BMI; Plicometry	+
						Body image	BIDA	+
						Perceived health	SF-12	+
Arkkukangas et al. (2020) [39]	Overall range: 60–88	F = 14.3%; M = 85.7%	J: Novices	Healthy and clinical	3; 1; 45–60	Physical performance	SPPB	+
						Self-efficacy	FES-S	+
Borba-Pinheiro et al. (2020) [56]	J: >40	NA	J: Novices	Healthy	NR	NA	NA	NA
Campos-Mesa et al. (2020) [40]	J: 74.3 ± 6.5; C: 77.8 ± 5.4	J: F = 21.1%; M = 79.9%; C: F = 100%; M = 0%	J: Novices	Healthy	6; 2; 60	Fear of falling	FES-I	+
Ciaccioni et al. (2020) [41]	J: 69.3 ± 3.9; C: 70.1 ± 4.5	J: F = 50%; M = 50%; C: F = 36%; M = 74%	J: Novices	Healthy	4; 2; 60	Walking patterns	Gait analysis	+
Franchini et al. (2020) [51]	J: 52.4 ± 2.4 (range = 50–59)	F = 0%; M = 100%	J: Intermediates and experts	Healthy	6; NR; NR	Handgrip strength	Dynamometry	+
						Anthropometry	Scale, stadiometer	+
Ciaccioni et al. (2021) [42]	J: 68.9 ± 3.7	F = 50%; M = 50%	J: Novices	Healthy	4; 2; 60	Training enjoyment	10-point scale	+
						Fear of falling	VAS scale	+
						Motivation	MPAM-R	+
						Self-regulation	SRQ-E	+
						Fall performance	Video analysis	+
Toronjo-Hornillo et al. (2021) [57]	Older people	NA	NA	Healthy	NR	NA	NA	NA
Sakuyama et al. (2021) [43]	J1:70 (range = 45–81); J2:72 (range = 62–83)	F = 86.8%; M = 13.2%	NR	Healthy	9; NA; 60	Quality of life	SF-36	?
Kujach (2022) [44]	J: 67.5 ± 5.3; C: 67.6 ± 5.1	F = 82.5%; M = 17.5%	J: Novices	Healthy	3; 3; 45	Anthropometry	Impedance analysis	?
						Cognition	Test form S8	+
						BDNF	Blood analysis	+
						Knee strength	Dynamometry	+

Note: NA = not applicable; NR = not reported. ^a^ Participants: age: when available, age (years) is reported as mean ± standard deviation values and range = min–max. Sex: F = females; M = males; Groups: C = control group; J = judo group; M = martial artists; S = sportive individuals. ^b^ Judo training load (*n*): months; sessions/week; min/session ^c^ Variables: BDNF = brain-derived neurotrophic factor; BMD = bone mineral density. ^d^ Measurement: DXA = dual energy X-ray absorptiometry; FES-I = Falls Efficacy Scale—International; FES-S = Falls Efficacy Scale—Swedish version; MPAM-R = Motives for Physical Activity Measure—Revised; *n* = number(s); OPAQ = Osteoporosis Assessment Questionnaire; SF-36 = Short Form-36 questionnaire; SRQ-E = Exercise Self-Regulation Questionnaire; SMTQ = Sports Mental Toughness Questionnaire; Sport-MPS2 = Sport-Multidimensional Perfectionism Scale 2; SPPB = Short Physical Performance Battery; VAS = Visual Analogue Scale. ^e^ Outcome: +positive (improvement); −negative (reduction); ? mixed.

### 3.4. Research Area, Variables, and Measurement Tools

From the experimental, observational, and methodological studies, three main research areas emerged: (i) health (13 studies, 56%), (ii) functional fitness (10 studies, 43%), and (iii) psychosocial aspects (10 studies, 43%). Regarding the reported variables and measurements (Table 2), health aspects (*n* = 5 studies) included bone mineral density, osteoporosis, anthropometry, rate of return to judo after surgery, and quality of life; the functional fitness aspects (*n* = 8 studies) encompassed balance, postural control, flexibility, strength (e.g., handgrip, and lower and upper body), coordination, physical performance, walking patterns, and falling performance; and the psychosocial aspects (*n* = 12 studies) included fear of falling, cognitive abilities, visual acuity, mental toughness, perfectionism, body image, perceived health, self-efficacy, training enjoyment, motivation, self-regulation, and the brain-derived neurotrophic factor. Regarding measurements and tools, nine studies reported device-based measures (e.g., X-ray absorptiometry, force plate, computerised tests, inertial sensors, plicometry, dynamometry, and gait and video analyses), nine studies applied self-reported measurement tools (e.g., questionnaires, surveys, and visual-analogue scales), and two studies used functional performance fitness tests under supervision of an external evaluator.

### 3.5. Outcomes

Table 2 reports the outcomes of judo training. The majority of the experimental studies (*n* = 9, 90%) indicated positive aspects, including bone- [35,36], functional fitness- [35,38,39,41,42,44], mental health-, and quality of life-related [37,38,39,42,43,44] aspects. One study [43] reported a mixed outcome of health-related quality of life, with the group of participants with lower levels of movement abilities at the baseline improving their physical and social functioning aspects, whereas the group with higher levels of movement abilities at the baseline improved in terms of their mental health-related aspects only. Using an impedance analysis, one study [44] reported no difference in anthropometric values after the judo program. However, the same study showed a combination of positive effects with respect to strength, balance, and cognition. When reported, adherence to the judo programs [38,41] was high (>65%). Adverse events were reported as null [39] or very low (<1%), including lumbago and meralgia [41], not requiring hospitalisation or treatment. Reasons for withdrawal or drop-out were diseases and other personal issues not pertinent with the research [41,44] or not specified [39].

Regarding the cross-sectional studies, 86% of the contributions indicated positive effects of judo on mental aspects [50], visual acuity [48,49], fitness and posture [47,51], and training capacity [45] when returning to adapted judo training (i.e., avoiding competitions and favouring randori and free practice) after joint replacement (76% of cases). Conversely, an observational study [46] found that age-related differences in postural control were larger in non-active individuals compared with a martial arts group, even though the two groups showed comparable sensorimotor and cognitive functions.

### 3.6. Study Quality

Table 1 summarises the evaluation of risk of bias for the included studies, while the detailed assessment of each item is reported in Appendix A. For non-randomised controlled trials, the risk of bias due to confounders resulted in being serious and moderate for 60% and 40% of the experimental studies, respectively. All experimental studies presented a low risk in the selection of the participants and the classification of the presented intervention. Only one paper [39] presented a moderate risk of deviation from the intended intervention, whereas this risk was low in the remaining 90% of the studies. No information on missing data resulted in 60% of the studies [35,36,37,39,40,41,42,43,44], with a low risk of missing data emerging in 40% of the contributions [38,41,42,44]. Whilst only one study [44] presented a low risk of bias in the measurement of the outcomes, 90% of the experimental studies presented a moderate risk. Finally, all the experimental studies presented a moderate risk of bias in the selection of the reported results.

For the cross-sectional studies, all the articles clearly specified and defined the research questions, objectives, and target population. Whilst two studies [45,47] did not report the participation rate of eligible individuals, for 71% of the observational studies, a ≥50% value for the participation rate of eligible persons emerged. A mixed picture emerged for eligibility criteria, with only one study [45] presenting uniform criteria, whereas this aspect was not present or impossible to determine in 57% and 29% of the studies, respectively. All contributions failed to include a power analysis. The study length was sufficient to elicit a number of outcomes in 86% of the studies, whereas for only one study [51], it was not possible to determine whether judo training exposure was long enough to elicit effects. Different judo programs were considered, with the specific aspects of judo training always being clearly defined, valid, reliable, and implemented consistently across all participants, even though the evaluations were performed only once, with 86% of the studies using valid and reliable outcome measurements. No blinded evaluation procedure was reported, with only one study [50] partially measuring and adjusting for key potential confounding variables.

For the methodological studies, all contributions described: the study objective; the target population; the inclusion of stakeholders in the developmental process; the methods for formulating the recommendations; the consideration of health benefits, side effects, and risks in formulating the recommendations; the explicit link between recommendations and supporting evidence; and specific and unambiguous recommendations. Conversely, a lack of procedures for updating the guidelines and discussion of potential organisational barriers in applying the recommendations emerged. Most of the studies [52,53,54,55,56] (83%) provided a specific description of the questions covered in the guidelines, piloted the guidelines among end users, and allowed for an easy identification of the key recommendations. Furthermore, 50% of the studies [54,56,57] sought participants’ views and preferences, providing clear criteria for selecting evidence. Only two studies [36,54] provided a clear definition of target users of the guidelines, used systematic methods to search for evidence, and presented key review criteria for monitoring and audit purposes. Only for 67% of studies [52,53,55,56] were the guidelines reviewed by external experts before publication, whereas 33% of studies did not present different options for the management of judo training [37,56] and did not support the guideline with tools for application [54,57]. Only one study [55] considered the potential cost implications of applying the recommendations, with 50% of the studies [52,55,56] being editorially independent from the funding body and reporting no conflicts of interest.

### 3.7. Novel Framework Including Recommended Exercises for Older Judoka

Figure 2 shows an overview of the applied guidelines issued from the methodological studies [52,53,54,55,56,57]. Health and risks emerged as the most frequent keywords connected to judo training, with social health appearing in all the studies, followed by quality of life and active aging (67%), physical health (50%), and mental health (33%). Injuries (100% of the studies) and falling (83%) were the most cited risks. One study53 warned that new disabilities could emerge from frailty and incipient functional limitations. Viable areas of intervention were condensed in two fields: (i) training methods (e.g., training principles and load, suggested exams) and exercises (e.g., specific techniques), and (ii) contextual barriers (e.g., fears, fragilities, environment) and facilitators (e.g., motivation, coach expertise, transgenerational relations).

## 4. Discussion

The main results of this review highlighted a serious risk of bias for 70% of the experimental studies, and a “fair” quality for 100% and 67% of the observational and methodological studies, respectively. Overall, the results showed that judo training promotes a series of positive outcomes related to psychosocial, functional fitness, and health aspects. Coherently with previous systematic reviews [12,24,29], the majority of the experimental studies (90%) showed clear evidence of the positive effects of judo training on health and fitness with advancing age, including significant improvements in fall skills and quality of life. Whilst it is widely recognised that regular physical activity promotes general well-being over the course of life, recently, the COVID-19 pandemic revealed the vital importance of active lifestyles for both mental and physical health, especially for the older population [4]. To accelerate actions towards the United Nation’s 2030 Agenda for Sustainable Development [4] target of a 15% relative reduction in physical inactivity in society, the WHO [5] has provided a set of evidence-based policy recommendations to prevent falls prevention, to maintain an independent life, to reduce social isolation, to strengthen social links, and to improve psychosocial health in older people. In fact, exercise programs for older people should consider the fulfilment of both physical and psychological domains [7,19,58].

As a multicomponent combat sport, judo training represents an extraordinary example of a suitable exercise program for older novice practitioners [2,23,26,38,59]. In fact, judo sessions include coordination and balance exercises performed barefoot, as well as an active engagement in activities and experiences that help older individuals in developing and maintaining their strength (i.e., *nage-waza* with practitioners moving and often lifting the partner) and flexibility (i.e., *ju-no-kata* practice with stretching), which could help in contrasting major degenerative aging processes (e.g., sarcopenia, osteoporosis) [26,29,38]. The present findings confirm previous results on the perceptual and cognitive benefits of different physical exercises for older people, with older judoka presenting better dynamic visual acuity and peripheral vision with respect to co-aged sedentary counterparts [60]. In situational sports and martial arts requiring fast adjustments to unpredictable contextual situations such as those which occur in judo, practitioners are trained to develop specific skills and peripheral vision to anticipate or to react to several sport requirements (i.e., in free situation of *randori* or during *kata* practice or repetitive techniques, known as *uchi komi*) [26]. The present review considered also aspects related to the emotional domains of mental health (e.g., fear of falling, mental toughness, and perfectionism). Advancing age is usually associated with a decrease in physical activity, self-esteem, confidence, strength, and balance, with a fear of falling increasing the risk of future falls [42,61]. The findings support judo training for decreasing the practitioner’s fear of falling, improving their self-confidence, body image, and perceived health-related quality of life [37,38,39,40,42]. A reduced fear of falling supports older individuals in both social activities (e.g., walking confidently in crowded places) and more difficult daily living tasks (e.g., walking on slippery surfaces) [37]. In older judoka, lower levels of concern regarding falling have been associated with a higher physical performance in the lower extremities, including strength, balance, and walking skills [39] and improved exercise enjoyment, competence, appearance, sociality, as well as intrinsic motivation [42]. According to the guidelines [61] recommending coaches to adopt a holistic approach to contextualise and assess falling risks in older people, judo training elicits a high fall competence associated with a high fall efficacy (i.e., *ukemi*, for the proper protection of head, elbows, and wrists hitting the floor during the fall) and a low fear of falling. In particular, judo coaches should be aware that sedentary individuals and novice judoka could have difficulties in performing a correct *ukemi*, which represents the predominant risk during the falls training and amplify the practitioner’s concerns of falling [37,42].

Judo aims to promote genuine and spontaneous-based movements shaped to the abilities of the training group [26], where collaboration and mutual learning are fundamental pillars of the relationships between practitioners [62]. Actually, judo showed to benefit the social interactions and self-esteem for a wide range of people, including resilient women in patriarchal contexts, individuals with disabilities, and, more recently, online judo practitioners confined at home during the COVID-19 lookdown [63,64]. Thus, it is not surprising that veteran athletes scored high for mental toughness and for striving for perfection with respect to elite and sub-elite athletes [50]. The practice of combat techniques in formal and choreographed approaches such as *kata* and in free modalities such as *randori* allow judoka self-expression and perfection by combining various physical exercises and traditional values (e.g., respect, courage, sincerity), which meet the necessities of older individuals [12,19,26]. During their lifelong practice, veteran judoka increase their skills, competences, and experience, which lead to deeper meanings and purposes to their sport with respect to health management and athletic victories [17,45,65].

This review also provides a novel overview of the suggested methodologies and practical and theoretical applications. Although there are proposed practical suggestions for judo training to prevent injuries [66,67], a limited literature emerges regarding guidelines for older practitioners. The included studies developed recommendations based on a sound theoretical background, which was validated with the contribution of several stakeholders. In general, a comprehensive judo program for older people should focus on physical, mental, social, quality of life, and active aging improvements, while decreasing risks of falls, injuries, and disabilities. Thus, the judo program operated on both training and context sides [61] must follow the principle of safety, personalised training load, training monitoring, and periodical evaluations of the practitioner’s functional fitness, athletic skills (e.g., belt colour examinations), and level of enjoyment [31]. Whilst specific types of safe exercises are suggested (e.g., general exercises, known as *taiso*, basic and prearranged forms of techniques, i.e., *kata*), practices involving opponents need a strict control (e.g., fighting as competition, known as shiai, and free practice, called *randori*) [26]. Additionally, specific judo breakfall techniques (*ukemi-waza*) and techniques from a standing position (*tachi-waza*) and on the ground (*ne-waza*) are recommend, whereas techniques used to grasp the opponent’s joint (*kansetsu-waza*) and techniques of strangulation (*shime-waza*) should be avoided or practiced on the kata modality only [38,40,55]. Finally, recommendations also consider contextual factors [5,61]. Important barriers that prevent participation, commitment, and effective results include several types of concerns and worries: fear of falling, fear of contact, shyness, and embarrassment. Other important issues to be considered are fragility (i.e., limited strength, endurance, and flexibility, as well as the presence of osteoporosis and osteopenia) and a status of loneliness that participants could experience. Support can emerge from personal, family, and social spheres highlighting the possible benefits of judo with advancing age [2,21]. To motivate the practitioners, recognise their competence, autonomy, and relatedness, judo coaches should possess specific hard (e.g., judo knowledge) and soft (e.g., communication, problem solving) skills [10,15,42]. Furthermore, judo can become an extraordinary means of transgenerational connections, where young and older practitioners share their unique knowledge and abilities for a mutual enrichment [27,37,40,55]. Finally, an appropriate physical environment should be considered, with safe and clean training and resting rooms, and a dojo equipped with materials to respond to potential emergencies (e.g., ice packs, plasters, and defibrillators) [38,40,55].

In addressing three types of research designs (i.e., experimental, observational, and methodological) and synthesizing comprehensive health- and quality of life-related data, the present systematic review is particularly novel and provides useful information to both practitioners and researchers. However, some potential limitations need to be mentioned. Due to interventional, methodological, and statistical differences across the included studies, a quantitative meta-analysis was not possible. Moreover, the presence of the important risk of bias within the retrieved studies may weaken the generalisation of the present findings. Nevertheless, this review reported relevant information also for judo coaches for the development of sound training programs through a comprehensive appraisal of the characteristics of older judoka [27,31,61]. Future studies are needed within this research area, investigating the mechanisms through which judo training exerts its potentially positive effects in relation to personal, interpersonal, and societal ageing factors, particularly on human relations that have been indicated as key elements for senior citizens’ successful ageing [20].

## 5. Conclusions

Despite the fact that important sources of bias characterised the experimental studies and the “fair” quality of the observational and methodological studies, the present analysis of research designs, methods, practitioners, measures, and outcomes supports positive links between judo training and health, functional fitness, and psychosocial outcomes. In light of a continued professional development, the present systematic review was also committed to support coaches’ knowledge and the advanced methodology of teaching judo techniques to middle-aged and older practitioners. However, to strengthen or to refute the reported results, future studies should attempt to conduct a quantitative meta-analysis of the results. In conclusion, this work aims to promote the further engagement of the scientific community in research lines concerning judo during the life course.

## Figures and Tables

**Figure 1 sports-11-00068-f001:**
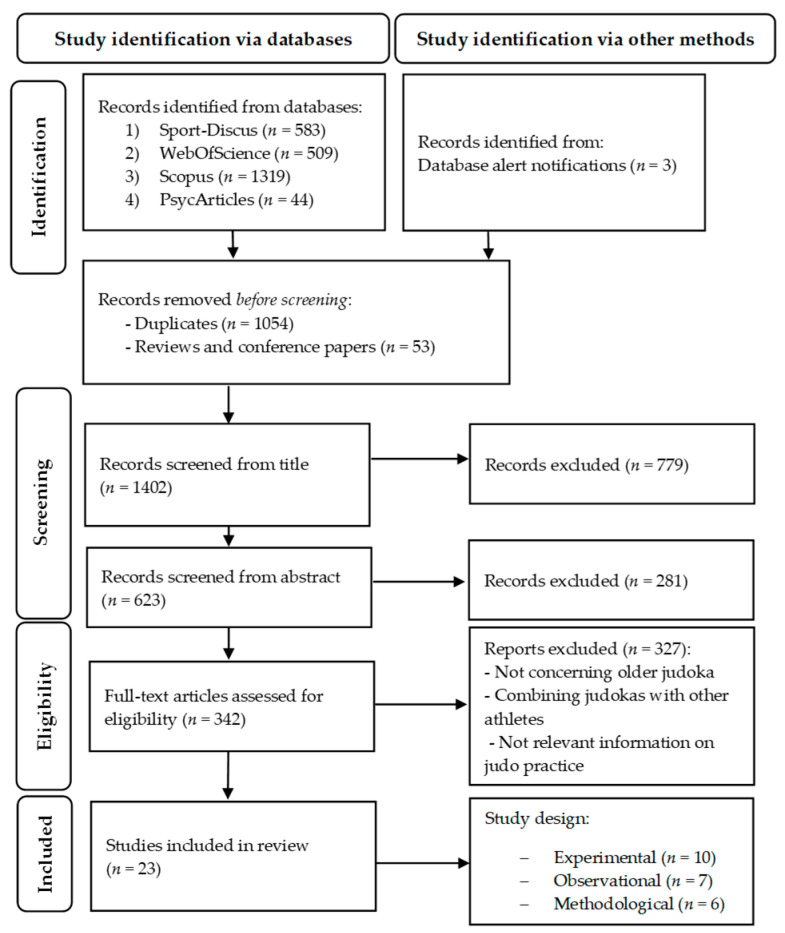
Flow chart of the systematic process of review.

**Figure 2 sports-11-00068-f002:**
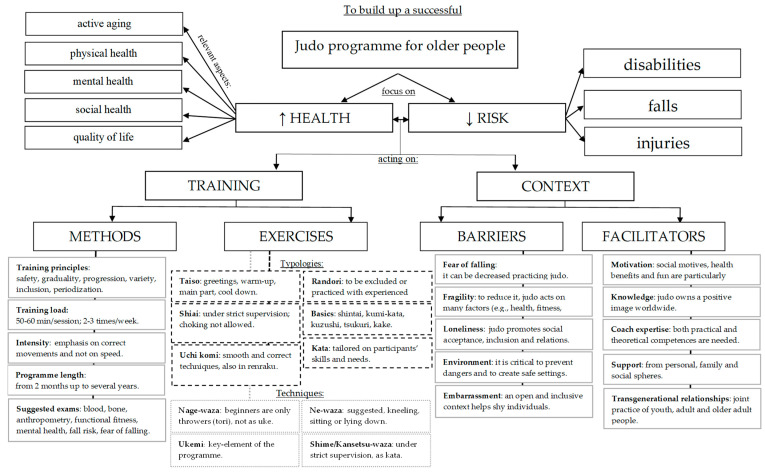
Overview of applied guidelines issued from the methodological studies [52,53,54,55,56,57].

## Data Availability

No new data were created.

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
