# Peer review of "Risks and Benefits of Judo Training for Middle-Aged and Older People: A Systematic Review"

_sports, 2023, doi:10.3390/sports11030068_

Round 1

Reviewer 1 Report

I have reviewed the manuscript titled “Judo for Older Adults: A Systematic Literature Review” and I provide some points for the authors to consider for improving their manuscript. The manuscript is well written and will be of interest to researchers and practitioners who support physical activity for older adults. I suggest a few minor revisions on the manuscript that will improve the manuscript.

1.       I suggest the authors consider the title of the manuscript as currently it is vague. It could be changed to incorporate the purpose of the review was to determine benefits and risks of judo training. The risks and benefits of Judo training for Older Adults: A Systematic Literature Review”

2.       I feel the introduction section is well written, however it would benefit from clarifying what age is deemed to be an “older adult”.

3.       The studies included within the review and then the subsequent outcomes seem to present a positive benefit of judo for older adults. However, there are some studies included within table 2 that report NA to variables, measurement and outcome (i.e., Borba-Pinheiro et al. (2013b) ; Campos-Mesa et al. (2015) etc.). These studies are not the methodological studies [52-57], therefore, it’s not clear how they passed the inclusion criteria for the systematic review.  

Author Response

Reviewer 1

Comments and Suggestions for Authors

I have reviewed the manuscript titled “Judo for Older Adults: A Systematic Literature Review” and I provide some points for the authors to consider for improving their manuscript. The manuscript is well written and will be of interest to researchers and practitioners who support physical activity for older adults. I suggest a few minor revisions on the manuscript that will improve the manuscript.

According to this comment, we deeply appreciate the opinion of Reviewer 1.

  1. I suggest the authors consider the title of the manuscript as currently it is vagu It could be changed to incorporate the purpose of the review was to determine benefits and risks of judo training. The risks and benefits of Judo training for Older Adults: A Systematic Literature Review”

We thank Reviewer 1 for the opportunity to improve the title. To comply also with the requests of Reviewer 2, the title has been revised as “Risks and Benefits of Judo Training for Middle-Aged and Older People: A Systematic Review”.

  1. I feel the introduction section is well written, however it would benefit from clarifying what age is deemed to be an “older adult”.

We thank Reviewer 1 for this feedback. We used the definition of W. Spirduso et al., 2005 [31]. To further clarify this aspect in the Method sections, the Eligibility Criteria have been revised as follows: “ii) studies involving Middle-Aged (i.e., ≥45-65 years) and Older (i.e., ≥65 years) practitioners [31]”

  1. The studies included within the review and then the subsequent outcomes seem to present a positive benefit of judo for older adults. However, there are some studies included within table 2 that report NA to variables, measurement and outcome (i.e., Borba-Pinheiro et al. (2013b) ; Campos-Mesa et al. (2015) etc.). These studies are not the methodological studies [52-57], therefore, it’s not clear how they passed the inclusion criteria for the systematic review.

We deeply thank Reviewer 1 for this feedback. Coherently, we implemented the manuscript, Table 1, Table 2, Figure 2, and Supplementary Material, changing the references’ codes. The tables have been now emended according to the order of the references. Accordingly, the studies reporting the labels NA are the methodological studies.

Reviewer 2 Report

I thank the authors for the good review they have done on an emerging topic that provides new opportunities for older people. Here are some suggestions to improve the presentation on the systematic review:

- I suggest change A systematic literature review x A Systematic Review

- I suggest change the title and aim to older adults x middle-aged and older people, since the average age considered is 45 years and over. Fact that they must also adapt in the methodology to be consistent.

- Change older adults x older people throughout the manuscript.

- I suggest incorporating a paragraph (prior to the aim) that justifies the need to carry out the systematic review.

- The flowchart must be presented in the results (study selection section) and explained here!

- The PICOS criteria could expose them in a table and they should be consistent with the inclusion and exclusion criteria (in fact those should be).

- I suggest adding a paragraph indicating the data synthesis before the results.

- In table 2 I suggest adjusting the meaning of the symbols: +: improve; -: reduction.

- I suggest adding a section at the end of the results regarding adherence and adverse events reported by the studies.

- It is necessary to add a paragraph of limitation and strengths of the systematic review at the end of the discussion.

- In the conclusion they should remove the citations and refere only to the findings of their review. In the same way I suggest reviewing the wording to be more purposeful, although I agree on the positive effects, without meta.-analysis it is not possible to provide conclusive information!!! 

Author Response

Reviewer 2

Comments and Suggestions for Authors

I thank the authors for the good review they have done on an emerging topic that provides new opportunities for older people.

According to this comment, we deeply appreciate the opinion of the reviewer 2.

Here are some suggestions to improve the presentation on the systematic review:

- I suggest change A systematic literature review x A Systematic Review

- I suggest change the title and aim to older adults x middle-aged and older people, since the average age considered is 45 years and over. Fact that they must also adapt in the methodology to be consistent.

We thank Reviewer 2 for the opportunity to improve the title. To comply also with the requests of Reviewer 1, the title has been revised as “Risks and Benefits of Judo Training for Middle-Aged and Older People: A Systematic Review”.           

- Change older adults x older people throughout the manuscript.

To comply with this suggestion, we changed the term “adults” with “people”.

- I suggest incorporating a paragraph (prior to the aim) that justifies the need to carry out the systematic review.

As suggested, we incorporated a sentence highlighting the rationale for this systematic review, as follows: “Thus, to cover this knowledge gap in the understanding of physical, mental, and social effects and correlates of judo training in older practitioners, there is a need to collect comprehensive information from both a theoretical and a practical perspective to guide the implementation of effective and safe training programs for older people.

- The flowchart must be presented in the results (study selection section) and explained here!

As suggested, the flowchart has been moved and explained in the Results within the Study Selection section.

- The PICOS criteria could expose them in a table and they should be consistent with the inclusion and exclusion criteria (in fact those should be).

We thank Reviewer 2 for the feedback. To avoid repetitions, we preferred to leave PICOS criteria in the manuscript text.

- I suggest adding a paragraph indicating the data synthesis before the results.

We thank Reviewer 2 for the very useful feedback. To guarantee the flow of the article, we incorporated the data synthesis part in the 2.4 section, which is now named “Data Extraction and Synthesis, with the following paragraph “According to the PRISMA guidelines [30], data were processed through a qualitative synthesis. Effects and correlates of judo training were considered in relation to participants’ chronological age and activity group (i.e., sedentary, judoka, sportive controls). Then, detailed tables and figures reporting major characteristics and findings of the selected studies were created”.

- In table 2 I suggest adjusting the meaning of the symbols: +: improve; -: reduction.

As suggested, we modified the caption note as follows: “+ positive (improvement); - negative (reduction); ? mixed”.

- I suggest adding a section at the end of the results regarding adherence and adverse events reported by the studies.

As suggested, we added at the end of results of the experimental studies the following paragraph: “When reported, adherence to the judo programs [38, 41] was high (>65%). Adverse events were reported as null [39] or very low (<1%), including lumbago and meralgia [41] not requiring hospitalization or treatment. Reasons for withdrawal or drop-out were diseases and other personal issues not pertinent with the research [41, 44] or not specified [39].

- It is necessary to add a paragraph of limitation and strengths of the systematic review at the end of the discussion.

- In the conclusion they should remove the citations and refer only to the findings of their review. In the same way I suggest reviewing the wording to be more purposeful, although I agree on the positive effects, without meta.-analysis it is not possible to provide conclusive information!!! 

As suggested, a paragraph addressing limitations and strengths has been added at the end of the discussion. Furthermore, we removed the part containing citations not referring to the review findings moving it at the end of the discussion section. To comply also with the suggestion of Reviewer 3, the conclusion now appears more streamlined as follows: “In addressing three types of research designs (i.e., experimental, observational, and methodological) and synthesizing comprehensive health- and quality of life-related data, the present systematic review is particularly novel and can provide useful information to both practitioners and researchers. However, some potential limitations need to be mentioned. Due to interventional, methodological, and statistical differences across the included studies, a quantitative meta-analysis was not possible. Moreover, the presence of important risk of bias within the retrieved studies may weaken the generalization of the present findings. Nevertheless, this review reported relevant information also for judo coaches for the development of sound training programs through a comprehensive appraisal of the characteristics of older judoka [27,31,61]. Future studies are needed within this research area, investigating the mechanisms through which judo training exerts its potentially positive effects in relation to personal, interpersonal, and societal ageing factors, particularly on human relations that have been indicated as key-elements for the senior citizens’ successful ageing [20].”

Reviewer 3 Report

Dear Authors

You have written a comprehensive review focused on studies investigating the benefits and risks of judo 11 training in older adults.

The abstract is ok.

Introduction

Judo is a combat sport and a martial art. Therefore, please use those two terms jointly throughout the text.

Methods:

Eligibility criteria are sufficient

Database search progress - why wasn't the PubMed database included?

Why was not the keyword "recreational" included in the search?

Describe what this means by authors specialised in judo (be specific for repeatability purposes).

Prisma flowchart included and correctly presented.

Quality assessment was done at a high level which adds to the robustness of this review.

Results are well presented.

The only thing I recommend adding is the information on judo training experience in identified 23 studies. 

Figure 2 is a great summary of the main findings. Well done

The discussion is well written. The only thing I see is out of focus is the importance of coaches' knowledge and advanced methodology of learning judo techniques to adult participants in the programs of continued professional development / CPD 

Conclusion - be specific

Line 400 - its effect - was it positive or negative? Be specific

Overall a very good review paper. Therefore, I recommend minor revisions.

Kind regards

Author Response

Reviewer 3

Comments and Suggestions for Authors

Dear Authors

You have written a comprehensive review focused on studies investigating the benefits and risks of judo 11 training in older adults.

The abstract is ok.

According to this comment, we deeply appreciate the opinion of reviewer 3.

Introduction

Judo is a combat sport and a martial art. Therefore, please use those two terms jointly throughout the text.

According to this comment, we emended the discussion section highlighting the dual nature of judo as follows: “In situational sports and martial arts requiring fast adjustments to unpredictable contextual situation as judo […]”.

Methods: Eligibility criteria are sufficient

We thank Reviewer 3 for this feedback.

Database search progress - why wasn't the PubMed database included?

We thank Reviewer 3 for the question. With filters based on our inclusion (e.g., humans, clinical and observational studies) and exclusion criteria (e.g., review, meta-analysis), a search piloting on PubMed resulted in a lower size (370 hits) compared to the other databased except PsychArticles. Nevertheless, we double-checked every single study and we did not find any further relevant article.

Why was not the keyword "recreational" included in the search?

We thank again Reviewer 3 for this valuable suggestion. According to this comment, we have now tried to add the keyword “recreational” in our search. This resulted in a slightly higher number of results. However, no further relevant article has been retrieved.

Describe what this means by authors specialised in judo (be specific for repeatability purposes).

According to this comment, we described the specializations as follows: “(i.e., certified kinesiologists and 4th dan black belt)”.

Prisma flowchart included and correctly presented. Quality assessment was done at a high level which adds to the robustness of this review. Results are well presented. The only thing I recommend adding is the information on judo training experience in identified 23 studies.

We thank Reviewer 3 for these valuable feedbacks. We tried to provide a consistent definition of “judo training experience” reporting the following paragraph in the Results section: “Regarding the judo background of participants, 13 studies focused on novices, four on high level judoka, four on amateur or intermediate level judoka, and three studies did not report this information.”  However, it wasn’t possible to retrieve more accurate data.

Figure 2 is a great summary of the main findings. Well done

We thank the Reviewer 3 for this useful feedback.

The discussion is well written. The only thing I see is out of focus is the importance of coaches' knowledge and advanced methodology of learning judo techniques to adult participants in the programs of continued professional development / CPD

According to this comment, we added a paragraph in the conclusion section highlighting the valuable focus on CPD: “In light of a continued professional development, the present systematic review was also committed to support the coaches' knowledge and advanced methodology of learning judo techniques to middle-aged and older practitioners.”

Conclusion - be specific

According to this comment, we modified the Conclusion section as follows: “Despite important sources of bias characterized the experimental studies and the “fair” quality of the observational and methodological studies, the present analysis of research designs, methods, practitioners, measures and outcomes supports positive links between judo training and health, functional fitness and psychosocial outcomes. In light of a continued professional development, the present systematic review was also committed to support the coaches' knowledge and advanced methodology of learning judo techniques to middle-aged and older practitioners. However, to strengthen or to refute the reported results, future studies should try also to apply a quantitative meta-analysis of the results. In conclusion, this work could promote a further engagement of the scientific community in research lines on judo during the life course.

Line 400 - its effect - was it positive or negative? Be specific

According to this comment, we modified the text as follows: “Future studies are needed within this research area, investigating the mechanisms through which judo training exerts its potentially positive effects in relation to personal, interpersonal, and societal ageing factors, particularly on human relations that have been indicated as key-elements for the senior citizens’ successful ageing [20].

Overall a very good review paper. Therefore, I recommend acceptance after minor revisions.

We have really appreciated the feedbacks received from Reviewer 3 and we would like therefore to thanks him/her once again.

Round 2

Reviewer 2 Report

I thank the authors for addressing all comments. The new version of the article is ready to be published!!!